# Is spatial exposure to heritage associated with visits to heritage and to mental health? A cross-sectional study using data from the UK Household Longitudinal Study (UKHLS)

Laura Macdonald [1], Natalie Nicholls [1], Eirini Gallou [2], Linda Monckton,[2] Richard Mitchell [1]

[1]MRC/CSO Social and Public Health Sciences Unit, University of Glasgow, Glasgow, UK
[2]Policy and Evidence, Historic England, London, UK

**Correspondence to**
Mrs Laura Macdonald;
Laura.Macdonald@glasgow.ac.uk

## ABSTRACT

**Objectives** Existing research highlights the beneficial nature of heritage engagement for mental health, but engagement varies geographically and socially, and few studies explore spatial exposure (ie, geographic availability) to heritage and heritage visits. Our research questions were 'does spatial exposure to heritage vary by area income deprivation?', 'is spatial exposure to heritage linked to visiting heritage?' and 'is spatial exposure to heritage linked to mental health?'. Additionally, we explored whether local heritage is associated with mental health regardless of the presence of green space.

**Design** Data were collected from January 2014 to June 2015 via the UK Household Longitudinal Study (UKHLS) wave 5. Our study is cross-sectional.

**Setting** UKHLS data were either collected via face-to-face interview or online questionnaire.

**Participants** 30 431 adults (16+ years) (13 676 males, 16 755 females). Participants geocoded to Lower Super Output Area (LSOA) 'neighbourhood' and 'English Index of Multiple Deprivation' 2015 income score.

**Main exposures/outcome measures** LSOA-level heritage exposure and green space exposure (ie, population and area densities); heritage site visit in the past year (outcome, binary: no, yes); mental distress (outcome, General Health Questionnaire-12, binary: less distressed 0–3, more distressed 4+).

**Results** Heritage varied by deprivation, the most deprived areas (income quintile (Q)1: 1.8) had fewer sites per 1000 population than the least deprived (Q5: 11.1) (p<0.01). Compared with those with no LSOA-level heritage, those with heritage exposure were more likely to have visited a heritage site in the past year (OR: 1.12 (95% CI 1.03 to 1.22)) (p<0.01). Among those with heritage exposure, visitors to heritage had a lower predicted probability of distress (0.171 (95% CI 0.162 to 0.179)) than non-visitors (0.238 (95% CI 0.225 to 0.252)) (p<0.001).

**Conclusions** Our research contributes to evidence for the well-being benefits of heritage and is highly relevant to the government's levelling-up heritage strategy. Our findings can feed into schemes to tackle inequality in heritage exposure to improve both heritage engagement and mental health.

## STRENGTHS AND LIMITATIONS OF THIS STUDY

⇒ Our paper is the first to explore associations between spatial exposure to heritage, and visits to heritage, and mental health, within relatively small areas across England.

⇒ We use a sample of adults from across England, with sampling weighted to ensure representativeness to the wider population of England.

⇒ We use a variety of Lower Super Output Area-level measures to capture exposure to heritage, that is, population and area densities, and percentage of sites as 'at risk'.

⇒ We explore whether associations between mental health and heritage remained regardless of the presence of local green space, providing a unique strand to this research.

⇒ Our study was cross-sectional; thus, we cannot assume that key variables demonstrate causal associations.

## INTRODUCTION

Population health research has paid much attention to the health benefits of exposure to natural environments.[1] Regular contact with green space has been associated with better physical health, for example, through reduced risk of hypertension, stroke, cardiovascular disease and asthma, and improved immune function.[2] There is stronger evidence for mental health benefits of time spent in green and natural space, such as improved cognitive function and brain activity,[1] greater social cohesion, and sense of community, relaxation and restorative psychological effects, and lower levels of depression, anxiety and stress.[3]

A much smaller evidence base suggests that visiting heritage may also benefit mental health and well-being. In this study, we define heritage as a building, monument,

site or area classified as having 'a degree of significance' worthy of consideration in planning-related decisions, due to its architectural or historic interest.[4] Heritage assets may be 'designated' by the Secretary of State for Digital, Culture, Media and Sport (DCMS) as advised by Historic England, or recognised for their significance by local planning authorities, while 'World heritage sites' are further recognised for their international value by bodies such as UNESCO. People who frequently visit heritage sites report higher life satisfaction,[5–7] enhanced levels of self-esteem and happiness[8] and lower levels of mental distress.[9] Residents of places with greater numbers of heritage sites have a stronger 'sense of place'[10] and higher levels of place attachment have been associated with better quality of life[11] and 'social well-being'.[12] Studies have also shown that engaging with heritage can be psychologically restorative,[13] and that visits to cultural heritage sites can reduce stress.[14] The denial of access to heritage during the COVID-19 lockdowns also provided evidence for its benefits. Awareness of the value of heritage to mental well-being was reportedly reinforced by being unable to access it.[15]

One problem with studies of both green space and heritage is that they do not often acknowledge the overlap between them, that is, the potential for co-occurrence of green space and heritage in the same area. Historic parks and gardens are 'cultural assets' in that they are commonly designed or created by and for people with a purpose. They are often cultural landscapes in their own right, but many parks and gardens contain historic buildings or have historic value, and many monuments and buildings are close to or surrounded by green spaces. Furthermore, the parts of towns and cities that contain more (or less) heritage might well also contain more (or less) green space and trying to 'untangle' the contributions of each environment to health and well-being is a difficult task. Overall, though, evidence suggests that exposure to heritage has the potential to benefit health, and in particular, mental health.

Despite the potential benefits, visits to heritage are unequal. People living in more deprived areas are less likely to engage in the arts, culture and heritage regardless of personal socioeconomic status.[9] In the UK in 2019/2020, a higher proportion of adults resident in the least deprived areas (83%) had visited sites in the past year than those resident in the most deprived (50%).[16] Heritage engagement levels vary by individual characteristics too with, for example, those in employment, and in higher socioeconomic groups, more likely to visit.[17] Geographical availability of cultural assets, such as museums and libraries is associated with visits or engagement,[18] but availability is unequal across the UK with less deprived areas showing better access.[19] In this study, we refer to this geographical availability factor as 'exposure'. There is surprisingly little evidence about exposure to heritage, about how it interacts with the socioeconomic characteristics of the population and whether it is implicated in any health benefits. Such evidence is important

to both agencies concerned with improving and equalising population health, and those seeking to understand the value and potential of heritage. If levels of exposure make a difference for both engagement and health, there is potential for place-based schemes to improve and equalise it.

To address this gap, in this paper we explore whether spatial exposure to designated heritage (ie, listed buildings, scheduled monuments and historic parks/gardens) varies by area income deprivation within England, and whether spatial exposure to heritage is linked to visits to heritage (unadjusted and adjusted for sociodemographic factors). We then explore whether exposure to heritage is associated with mental health among all residents, and those who are visitors and non-visitors (controlling for sociodemographic factors and other green space). Additionally, we explore 'built' heritage, specifically, to understand whether exposure to listed buildings and scheduled monuments is associated with mental health benefits beyond those associated with exposure to green space. We consider whether exposure to heritage in an area is related to psychological well-being among those living in proximity to it and disentangle the potential overlap in associations with well-being related to heritage exposure and to green space exposure. Our research is timely, with, in the 2022 Levelling Up paper, the UK government showing a commitment to tackling sociospatial inequalities in heritage access and engagement.[20]

In summary, our three research questions are: 'does spatial exposure to heritage (and heritage 'at risk') vary by area deprivation?', 'Is spatial exposure to heritage linked to visiting heritage?' and 'Is spatial exposure to heritage (ie, any, or 'built' heritage specifically) linked to mental health among heritage visitors and non-visitors?'.

## METHODS

To answer our research questions, we needed four data components describing: (i) local area population, context and socioeconomic situation; (ii) spatial exposure to heritage; (iii) spatial exposure to other green spaces and (iv) individual level visits to heritage, health and sociodemographic covariates. These data needed a common spatial framework to represent, and georeference individuals (ie, residential location related to a ground system of geographic coordinates) to, the 'local area' or 'neighbourhood'. The spatial framework was Lower Super Output Area (LSOA) (as individual addresses or postal codes not available). LSOAs are small area units created for collecting and reporting statistics about the UK population.[21] They are socially homogeneous and consistent in population size; however, their geographical size varies according to level of urban/rural (eg, rural LSOAs are geographically larger as their populations are more dispersed). LSOAs have a median area size of 0.46 km² and median population size of 1500 residents or 650 households (see online supplemental table 1 for more

information). There are 32 844 LSOAs in England and they are commonly used as a proxy for 'neighbourhood'.

## DATA

### Local area population, context and socioeconomic situation

We obtained the English Index of Multiple Deprivation (EIMD) 2015 (income domain) ranks and quintiles[22] and 2015 mid-year population estimates[23] to describe LSOA population and socioeconomic situation. We did not use the full EIMD as it includes barriers to local services and health variables, therefore, there may have been some circularity in investigating whether the full EIMD was associated with access to heritage/green space and/or mental health. An EIMD Income quintile descriptive is included within online supplemental table 1 (including total and mean population/size per quintile).

### Spatial exposure to heritage and other green spaces
#### Heritage data

We obtained spatial data on all nationally protected historic buildings and sites from Historic England for 2014.[4] The National Heritage List for England originated in 1882 when the first powers of protection were established and became a statutory 'Listing' after World War Two (mid-1940s). Currently, the list holds over 400 000 entries and draws together all scheduled monuments, listed buildings, registered landscapes, battlefields and protected wrecks. Historic England continuously updates the list, curated from the DCMS.[4] In our analysis, we included listed buildings (points only—in compliance with Ordnance Survey (OS) licence), scheduled monuments (polygons) and historic parks/gardens (polygons). We excluded wreck sites (not visible from land) and excluded battlefields as small numbers of these sites (n=47, sited within 0.1% of LSOAs only) would not allow meaningful comparison across income quintiles. Listed buildings are buildings (eg, residences, farmhouses, churches, etc) recognised as being of special architectural or historic interest. Scheduled monuments are historic sites of national importance that are legally protected, such as Roman remains, burial mounds, castles, bridges, earthworks, etc. Historic parks/gardens are protected 'designed' landscapes, including a range of planned open spaces, such as public parks, cemeteries, private house grounds, etc. The dataset also distinguished 'Heritage at Risk' (HAR) sites. HAR is an official description of a heritage asset added to the HAR register that is maintained by Historic England, and records the condition of designated heritage assets after assessment. Such sites are vulnerable to loss due to 'neglect, decay or inappropriate development' and are in need of safeguarding and protection.[24]

#### Other green space data

We obtained data on other green space for England from the OS Open Greenspace database. Data for 2014 were not available, with the closest temporal match being July 2017.[25] These were vector data (scale 1:25000) including polygon boundaries of green spaces. We selected green spaces that we considered 'natural heritage', including: allotments and community growing spaces; cemeteries and other religious grounds; golf courses; and parks/gardens. We excluded spaces associated with sport/play: tennis courts; bowling greens; other sports facilities; playing fields; and play spaces. Golf courses may be regarded as sports facilities; however, these were included as they are often geographically large and highly visual green spaces, which may contain publicly accessible footpaths and can be surrounded by other green space or coastal walks (ie, proximal access is available). Some heritage sites included in the data from Historic England were likely to also feature in the OS data. To ensure we did not double count these, we overlaid the two datasets in a Geographic Information System (GIS), ArcMap V.10.8.1, to check for and remove duplicates; any feature captured in the data from Historic England was removed from the OS green space data. The remaining features are referred to as 'other green space'. We obtained LSOA boundaries (2011)[26] and used the 'Overlay' 'Spatial Join' tool to link the points (or polygon centroids/centre points) in the heritage and other green spaces datasets to LSOA boundaries. If a site (polygon) was spread across more than one LSOA, it was linked to the LSOA in which its centroid was located. We then calculated a count of each heritage feature and a count of each other green space feature in each LSOA.

#### Exposure variables

We then used IBM SPSS statistics V.28 to calculate three categorical exposure measures of heritage (at LSOA level). These were 'presence or absence of heritage sites', 'heritage sites per 1000 population' and 'heritage sites per kilometre squared ($km^2$)'. Using these three measures allowed us to model whether certain 'levels of' heritage were important, and allowed us to account for variation in the geographical size of LSOAs. The per population and per area exposure variables were grouped into 'none', 'low-medium levels (1–4 sites)' and 'higher levels' (five or more sites). Due to no existing methodology on appropriate groupings of exposure we based our thresholds on the distribution of values and on percentile representations that is, numbers roughly represented none, 25th–50th percentile and 75th percentile, respectively. We also used the same method detailed above to create the per population and per area exposure variables for: 'built' heritage (ie, listed buildings/scheduled monuments), historic parks/gardens, other green space (non-heritage) and any green space (heritage and non-heritage combined). We joined the area measures of heritage, 'built' heritage, other green spaces, and any green spaces to individual level data via the individual LSOA of residence. UK Household Longitudinal Study (UKHLS) individuals' LSOA codes were accessed via a UK Data Service special licence.

## Individual-level visits to heritage, health and sociodemographic covariates

### Visits to heritage

We undertook a secondary analysis with individual-level data drawn from wave five (2014) of the UKHLS (2021) (see online supplemental table 2 for sample descriptive). Data were obtained via the UK Data Service. The UKHLS is a large panel survey, which used a clustered and stratified, probability sampling method, and tracked around 31 000 people in England since 2009 (for full details of the survey and its methodology including sampling methods see Kaminska and Lynn[27]).We used wave five as it was the most recent wave with questions about visits to heritage. Respondents were asked how often (ie, 'not once in the last 12 months', 'once in the last 12 months', 'two times in the last 12 months', 'at least three or four times a year', 'at least once a month', 'at least once a week') they visited each of these heritage sites: 'a city or town with historic character', 'a historic building', 'a historic place of worship', 'a historic park or garden', 'a place of industrial history or historic transport system', 'a monument such as a castle, fort or ruin', 'a site of archaeological interest' or a 'sports heritage site', in the past year. We did not know whether visits to heritage were in respondent's LSOA or not.

### Mental health measure and covariates

UKHLS carries a General Health Questionnaire-12 (GHQ-12) subjective well-being measure, which we included as a measure of mental health. The GHQ-12 includes 12 questions evaluating general mental health functioning and distress. Responses use a 4-point scoring system, that is, better than normal, same as usual, worse than usual, or much worse than usual, and responses are summed to a score varying from 0 to 12.[28] We modelled the score as a binary variable lower distress (0–3) and higher distress (ie, minor psychiatric morbidity) (4+); the range of GHQ-12 values was not appropriate for use as a continuous numeric outcome as it is a bounded interval scale, with values being of an integer format. GHQ has been clinically validated, is a screening tool, and the threshold of four deemed valid to represent minor psychiatric morbidity 'case-ness' for the UK population, as this low level protects sensitivity.[29] Additionally, it has been used previously in green space and mental health research.[30] Covariates were chosen based on their established associations with visits to heritage and/or mental health,[17 31] and examination of full model coefficients confirmed the appropriateness of inclusion of these variables within models. They were sex; age group (10-year intervals); EIMD income (quintiles); household composition ('single adult', 'single adult with children under 16', 'two adults or more', 'two adults (or more) with children under 16'); ethnicity ('White', 'Black, Asian and Minority Ethnic/mixed' (numbers did not permit more specific groupings)); highest education qualification ('degree/other higher degree', 'A-levels/GCSEs', 'other qualification', 'no qualification'); socioeconomic

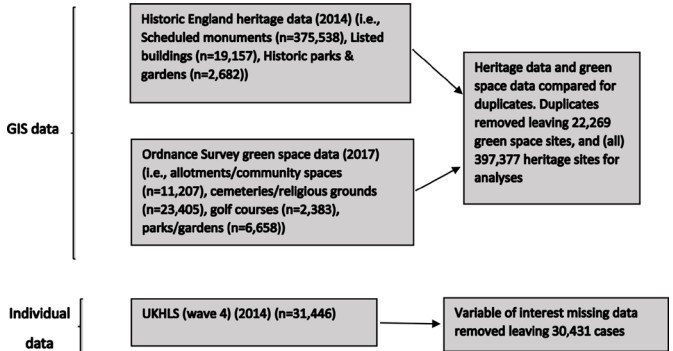

**Figure 1** Flow chart summary of GIS and individual data used. GIS, Geographic Information System; UKHLS, UK Household Longitudinal Study.

status (NS-SEC—'managerial/ professional', 'intermediate/small employment/own account', 'lower supervision/lower technician/semi-routine/routine', 'not in (paid) employment'); housing tenure ('owner occupier', 'social renter', 'private renter') and long-standing health condition/mobility ('none', 'condition without limited mobility', 'condition with limited mobility').

### Patient and public involvement

We did not involve study participants in the development of the research question, design and implementation of the study or interpretation of the results.

### Strengthening the Reporting of Observational Studies in Epidemiology

Strengthening the Reporting of Observational Studies in Epidemiology cross-sectional reporting guidelines were adhered to within this study.[32]

See figure 1 for a flow chart summary of GIS and individual data, including final numbers in analyses.

### Statistical analysis

To explore whether spatial exposure to heritage sites varies by deprivation, we compared mean/median densities of sites (per 1000 population, per square kilometre (km²)), across data zones within area income deprivation quintiles, using the Kruskal-Wallis test. This test was used as the distribution of these sites indicated that parametric comparisons were not appropriate. We also calculated percentage of heritage sites as 'at risk' across deprivation quintile.

To explore whether spatial exposure to heritage was linked to visiting heritage, we used multilevel logistic regression due to the inclusion of binary outcomes and random effects necessary due to sampling method. Regression was used to determine the association between heritage exposure measures (ie, presence of sites (none, 1+), sites per population (none, 1–4, 5+) and per area (none, 1–4, 5+)), and whether a respondent had visited a heritage site in the past year (ie, 'no visits', 'one or more visits yearly'), controlling for individual sociodemographic covariates (listed previously). We undertook further similar modelling to explore whether visits

to historic parks/gardens, and scheduled monuments, specifically, were related to exposure to these particular sites, respectively.

To explore whether spatial exposure to heritage was linked to mental health, we used multilevel logistic regression to determine the association between heritage exposure measures (as above) and the mental health measure (binary outcome). We controlled for sociodemographic variables, 'visits to heritage' and exposure to 'other green space'. Other green space was included as a confounder to account for potential associations between green space and mental health; such associations were established in prior research.[3] Initially, 'heritage exposure/visits' interactions were included in the models to investigate moderating effects on mental health, with different exposure type/visit interaction run in separate models. If on testing, an interaction was found non-significant, it was removed and models re-run with main effects only. Furthermore, we repeated models with 'built' heritage only (ie, listed buildings and scheduled monuments) and controlled for 'all green space' (ie, historic parks/gardens and other green space combined).

In all multilevel models, primary sampling unit (PSU) was set as random effect, weighted by the cross-sectional individual weights supplied by the data providers (see UKHLS[33] for further information on the PSU). Model results are presented as adjusted ORs and, for independent factors with more than two levels, predicted probabilities (of visits and of poorer mental health). In addition, post hoc multiple comparison testing was applied. Where applicable, to allow interpretation of results of a model's interaction term, predicted probabilities were presented, and subject to post hoc testing. Regarding missing data,

a complete case approach was used, that is, statistical analysis included only respondents for which there was no missing data on the variables of interest (n=30 431). Those included and excluded from analysis did not vary in terms of sex and education, but did somewhat for age, income and ethnicity, (included/excluded: 53%/54% females, 36%/36% degree or higher, mean age: 48/43 years, 21%/27% in most deprived areas, 82%/76% white). Multilevel models performed in STATA17,[34] with a global significance set at 5%.

## RESULTS

Heritage exposure varied by area deprivation in a manner dependent on the type of exposure (descriptive information on heritage numbers per quintile available within online supplemental table 1). Table 1 shows the mean/median distribution of heritage sites per population, and per area/km², across data zones, by EIMD income quintile. The most deprived areas (Q1) had lower numbers of heritage per 1000 population, while Q4 and Q5 had the highest numbers. This was evident for all three types of heritage analysed together and separately. The pattern for the numbers per area across income deprivation was less clear. The most deprived areas (Q1) had the lowest numbers of listed buildings (mean 9.08), scheduled monuments (0.06) and sites combined (9.23) (see table 1). Q2 had the highest number of listed buildings (12.92), and combined sites (13.10), while Q5 had the highest numbers of scheduled monuments (0.11). Though significant, there was very little variation in density of historic parks/gardens per area across the deprivation quintiles.

**Table 1** Mean/median distribution of heritage sites (2014) per 1000 population, and per area/km², by EIMD Income (2015) quintile

| EIMD income | Listed buildings mean/median (IQR) | | Historic parks and gardens mean/median (IQR) | | Scheduled monuments mean/median (IQR) | | Overall mean/median (IQR) | |
|---|---|---|---|---|---|---|---|---|
| | Per 1000 people | Per km² | Per 1000 people | Per km² | Per 1000 people | Per km² | Per 1000 people | Per km² |
| 1 (most deprived) | 1.79/0.00 (0.00–0.89) | 9.08/0.00 (0.00–4.61) | 0.02/0.00 (0.00–0.00) | 0.09/0.00 (0.00–0.00) | 0.03/0.00 (0.00–0.00) | 0.06/0.00 (0.00–0.00) | 1.83/0.00 (0.00–0.99) | 9.23/0.00 (0.00–4.80) |
| 2 | 3.73/0.48 (0.00–2.09) | 12.92/0.56 (0.00–6.28) | 0.03/0.00 (0.00–0.00) | 0.09/0.00 (0.00–0.00) | 0.08/0.00 (0.00–0.00) | 0.09/0.00 (0.00–0.00) | 3.84/0.52 (0.00–2.21) | 13.10/0.73 (0.00–6.53) |
| 3 | 6.80/0.73 (0.00–4.97) | 12.27/1.37 (0.00–5.91) | 0.04/0.00 (0.00–0.00) | 0.09/0.00 (0.00–0.00) | 0.37/0.00 (0.00–0.00) | 0.10/0.00 (0.00–0.00) | 7.21/0.79 (0.00–5.23) | 12.47/1.55 (0.00–6.11) |
| 4 | 11.39/1.57 (0.00–11.19) | 9.58/1.72 (0.00–4.56) | 0.07/0.00 (0.00–0.00) | 0.09/0.00 (0.00–0.00) | 0.72/0.00 (0.00–0.00) | 0.09/0.00 (0.00–0.00) | 12.19/1.72 (0.00–11.82) | 9.75/1.89 (0.00–4.76) |
| 5 (least deprived) | 10.42/1.57 (0.00–9.86) | 9.18/1.62 (0.00–4.25) | 0.08/0.00 (0.00–0.00) | 0.10/0.00 (0.00–0.00) | 0.54/0.00 (0.00–0.00) | 0.11/0.00 (0.00–0.00) | 11.05/1.74 (0.00–10.35) | 9.38/1.78 (0.00–4.47) |
| England | 6.83/0.64 (0.00–4.19) | 10.61/1.13 (0.00–4.99) | 0.05/0.00 (0.00–0.00) | 0.09/0.00 (0.00–0.00) | 0.35/0.00 (0.00–0.00) | 0.09/0.00 (0.00–0.00) | 7.22/0.66 (0.00–4.41) | 10.79/1.31 (0.00–5.20) |
| Kruskal-Wallis H value (df) | H(4)=2001 p<0.001 | H(4)=788 p<0.001 | H(4)=560 p<0.001 | H(4)=560 p<0.001 | H(4)=1428 p<0.001 | H(4)=1430 p<0.001 | H(4)=2049 p<0.001 | H(4)=714 p<0.001 |

EIMD, English Index of Multiple Deprivation.

**Table 2** Percentage of heritage sites as 'Heritage at Risk' (2016) (HAR) by EIMD income (2015) quintile

| EIMD income | Percentage as HAR | | | |
| --- | --- | --- | --- | --- |
| | Listed buildings | Historic parks/gardens | Scheduled monuments | All sites at risk |
| 1 (most deprived) | 2.0 | 3.4 | 21.7 | 2.3 |
| 2 | 0.8 | 2.7 | 16.4 | 1.1 |
| 3 | 0.6 | 3.8 | 15.4 | 1.4 |
| 4 | 0.4 | 3.0 | 13.2 | 1.2 |
| 5 (least deprived) | 0.3 | 4.1 | 12.6 | 1.0 |
| England | 0.6 | 3.5 | 13.8 | 1.2 |

EIMD, English Index of Multiple Deprivation.

Table 2 shows the percentage of heritage sites at risk by EIMD income quintile. Around 1.2% (n=4832) of heritage sites (ie, 2098 listed buildings, 2639 scheduled monuments and 95 historic parks/gardens) were deemed as HAR. Over 2% in total were found in the most deprived fifth of areas, compared with 1% in Q5 (and between 1.1% and 1.4% in the other quintiles). The most deprived areas had the highest percentage of HAR sites across two of the three categories (2.0% of listed buildings, 21.7% of scheduled monuments). In general, the percentage of HAR listed buildings and scheduled monuments increased as areas became more income deprived. The least deprived areas (Q5) exhibited the highest percentage of historic parks/gardens as 'at risk', that is, 4.1% of sites compared with 3.4% of sites in Q1.

Of 30 431 respondents, 59.9% had at least one heritage site within their LSOA, and 61% had visited heritage in the past year (n=19 232). Table 3 shows the findings of the multilevel logistic regression analysis as adjusted odds of visiting heritage by exposure to heritage sites. Respondents with at least one heritage site of any type in their neighbourhood (LSOA) were significantly more likely to have visited a heritage site in the past year (OR 1.12 (95% CI 1.03 to 1.22)), than those with no sites

**Table 3** Adjusted odds of visiting heritage by measures of exposure to heritage

| | | Odds of visiting heritage | |
| --- | --- | --- | --- |
| | | OR (95% CI) | P value |
| Heritage (any) | None | 1.00 | 0.008 |
| | 1+ | 1.12 (1.03 to 1.22) | |
| Heritage (any) per 1000 population | None | 1.00 | 0.014 |
| | 1–4 | 1.10 (1.00 to 1.20) | |
| | 5+ | 1.16 (1.05 to 1.30) | |
| Heritage (any) per km² | None | 1.00 | 0.027 |
| | 1–4 | 1.11 (1.01 to 1.22) | |
| | 5+ | 1.14 (1.02 to 1.26) | |

Adjusted by household type, sex, age, income deprivation, ethnicity, education, socioeconomic status, housing tenure and health condition/mobility.

(p=0.008). Those with five or more heritage sites per 1000 population, or per km², in their LSOA were most likely to have visited a heritage site (OR 1.16 (95% CI 1.05 to 1.30) and 1.14 (95% CI 1.02 to 1.26), respectively). The presence of scheduled monuments within respondents' neighbourhoods did not increase the probability of visiting monuments specifically, however, with an increase in LSOA-level historic parks/gardens per area the likelihood of visiting such sites increased (OR 1.12 (95% CI 1.04 to 1.21), p<0.01) (results not shown).

When evaluating associations between mental health and exposure to heritage, interaction effects between heritage exposure and heritage visits were found to be associated with mental health. Table 4 presents the predicted probabilities of poorer mental health by heritage exposure and visits to heritage. Post hoc testing (with Šidák adjustments for multiple comparisons) found the following associations: among those with any heritage, visitors to heritage had a lower predicted probability of distress (0.171 (95% CI 0.162 to 0.179)) compared with non-visitors (0.238 (95% CI 0.225 to 0.252)) (Šidák adjusted p value <0.001). This was also true when considering different levels of exposure, that is, five or more sites per 1000 population (visitors: 0.165 (95% CI 0.152 to 0.177), non-visitors: 0.241 (95% CI 0.219 to 0.264) (Šidák p=0.008)), or per km² (visitors: 0.171 (95% CI 0.158 to 0.184), non-visitors: 0.246 (95% CI 0.226 to 0.267) (Šidák p=0.016)). We found similar results when 'built' heritage (ie, listed buildings/scheduled monuments only) was included as the exposure variable (controlling for any green space). Among those with one or more built heritage sites, or with five or more per 1000 population, visitors had a lower predicted probability of distress. In the former, visitors: 0.171 (95% CI 0.163 to 0.179), non-visitors: 0.237 (95% CI 0.224 to 0.251) (Šidák p=0.001) and, in the latter, visitors: 0.165 (95% CI 0.152 to 0.178), non-visitors: 0.242 (95% CI 0.219 to 0.265) (Šidák p=0.009) (results not shown). As main effects, there were no associations between heritage exposure and distress (GHQ-12), regardless of exposure measure used (p values: 0.055–0.224).

**Table 4** Predicted probabilities and 95% CI of poorer mental health (General Health Questionnaire-12) for the interaction effects of heritage exposure and visits to heritage

| | | Non-visitors (95% CI) | Visitors (95% CI) | Interaction p value |
|---|---|---|---|---|
| Heritage | None | 0.218 (0.204 to 0.233) | 0.184 (0.174 to 0.195) | 0.005 |
| | 1+ | 0.238 (0.225 to 0.252)* | 0.171 (0.162 to 0.179)* | |
| Heritage per 1000 population | None | 0.219 (0.205 to 0.234) | 0.185 (0.174 to 0.196) | 0.011 |
| | 1–4 | 0.235 (0.218 to 0.251) | 0.173 (0.163 to 0.184) | |
| | 5+ | 0.241 (0.219 to 0.264)** | 0.165 (0.152 to 0.177)** | |
| Heritage per area (1 km²) | None | 0.219 (0.205 to 0.234) | 0.185 (0.174 to 0.195) | 0.014 |
| | 1–4 | 0.230 (0.213 to 0.248) | 0.169 (0.159 to 0.179) | |
| | 5+ | 0.246 (0.226 to 0.267)*** | 0.171 (0.158 to 0.184)*** | |

Post hoc multiple comparisons: *difference $p<0.001$, **difference $p<0.01$, ***difference $p<0.05$.
Adjusted by household type, sex, age, income deprivation, ethnicity, education, socioeconomic status, housing tenure, health condition/mobility, exposure to other green space and visits to heritage.

## DISCUSSION

This is the first study to explore the spatial distribution of designated heritage sites across areas of varying deprivation in England, and associations between exposure and visits, and mental health. We found that, compared with wealthier areas, the poorest areas had fewer sites per population, and, of these sites, a higher proportion were deemed 'at risk'. Neighbourhood exposure to heritage was associated with visits to heritage; however, the physical presence of neighbourhood heritage, on its own, was not associated with residents' mental health. The combination of having heritage present in the neighbourhood *and* visiting heritage (whether within the neighbourhood or further afield), was associated with better mental health.

Regarding the social-spatial patterning of heritage, the distribution of heritage varied according to how heritage exposure was measured. The most deprived areas showed the lowest mean heritage sites *per population* (Q1: 1.83 per 1000), and the least deprived areas the highest (Q4: 12.2 per 1000, Q5: 11.1 per 1000), however, mean sites *per area* for the least (Q1: 9.2 per km²) and most deprived areas (Q5: 9.4 per km²) were similar. This apparent contradiction in findings between measures likely reflects wealthier areas being on average larger in size. In other words, they do have more heritage sites per population, but these are spread across larger geographical extents (LSOA size increase with increasing wealth, see online supplemental table 1). Indeed, greater distance to a local heritage site may be less of a spatial barrier to those on higher incomes, whom are more likely to own cars.[35] Little existing research investigates spatial distribution of heritage on a national scale. A US city-based study, found that higher income/educated neighbourhoods, had superior access to historical and heritage institutions.[36] The authors described spatial access to historical/heritage institutions as 'elitist' with unequal geographic distribution detrimental to poorer neighbourhoods. In our study, perhaps the spatial bias towards higher income areas is a result of heritage increasing overall area wealth. Indeed,

conservation area homes in England benefit from price premiums and greater annual price growth,[37] and those with the financial means may pay more to live in homes with historic features.[38] We also report that more heritage sites within poorer areas were 'at risk'. HAR sites are more likely to be in a state of disrepair, less aesthetically pleasing and potentially unsafe, therefore their higher prevalence in poorer neighbourhoods may result in residents being less likely to engage with heritage. Other existing features of poorer neighbourhoods could compound this, such as higher environmental 'incivility', for example, graffiti, litter, noise and air pollution,[39] and vacant and derelict land.[40] It is conceivable that HAR's greater presence within lower income areas (and greater numbers of heritage sites (overall) per population within wealthier areas), is a result of inequity in the geographical distribution of funding; the Levelling Up paper reports the need for investment in culture and heritage outside London, and within poorer areas.[41]

The finding that respondents were more likely to have visited a heritage site in the past year, if they were exposed to local heritage, is comparable to previous evidence on spatial access and attendance at museums or galleries in London.[18] Brook[18] found that attendance rate variation could not be explained solely by people's individual characteristics and better spatial access was an important contributory factor. Authors concluded that well-resourced locales could provide significant 'opportunity structures' for cultural engagement.[18] Certainly better spatial access to other types of environmental features is associated with higher usage, such as sports facilities[42] and green space.[43] We found that those with a higher level of historic parks/gardens locally were more likely to have visited this type of green space in the past year. We cannot say why greater exposure increased the likelihood of heritage visits more generally, or whether local exposure equates to higher local site usage in particular. However, it is feasible that nearby heritage could be a significant 'pull factor' as reduced time and effort may be needed for visits, or visibility of local heritage may

contribute to social norms and increase interest in visits, whether near or far.

In terms of a main effect, we found no association between heritage exposure and psychological distress. Associations between heritage engagement and better mental health and well-being are established,[5–7 9 13 17] however, few studies focus on the mental health benefits of spatial exposure to heritage specifically. The lack of association, in our study, could be attributed to use of proximity as a 'proxy' for engagement with local heritage, while, in reality, location of visits was unknown. Future surveys could gather data on how far people travel to engage with heritage. We acknowledge that proximity does not indicate use, or awareness of, local heritage. Residents may lack awareness of local neighbourhood features, such as green space[44] or general amenities,[45] and this could be true of local heritage. We did find an interaction effect, among those with heritage locally; visitors to heritage showed lower psychological distress. A European-based study looked at associations between neighbourhood green/blue space and mental health,[46] and found that although people in highly green/coastal areas experienced better mental health, this relationship did not remain when controlling for recreational visits to these areas. Visit frequency appeared to mediate associations between green/blue space and well-being. Authors reasoned that better mental health among those in the greenest/bluest areas could be a result of such environmental qualities encouraging visits. Similarly, we found that a combination of proximal heritage, and visits to heritage, was associated with better mental health; higher spatial exposure appeared to provide a pathway to encourage visits. Our findings contribute to discussions about which type of neighbourhood exposure is key in associations with mental health. Many neighbourhood effects' studies define 'exposure' as 'residential proximity', for example, green space.[47] It is valuable to examine both spatial exposure, and direct contact/engagement, when considering mental health. The beneficial association between greenspace and mental health has been well researched, with theories of benefit based on visualising the structure of greenspace, its colours and its landscapes, to engage the human brain, and work to reduce stress, restore attention, and enhance well-being.[48] Less is known about how 'bricks and mortar' heritage influences mental health. We found that, regardless of green space exposure, those with built heritage in their local area, who had visited heritage, showed lower distress. Historic buildings are considered to improve neighbourhood quality with older buildings creating feelings of grandeur and permanence,[38] and exposure to urban environments with historic features, has been perceived more restorative, attractive and interesting than those without.[49 50] Viewing heritage appears to engage the senses, and places considered 'special' may stimulate the brain cells that manage emotion.[51] Residents who live in areas with higher proportions of listed buildings, or who regard a building as 'special' within their local area, have a greater 'sense of place'[10] which, in turn, can be associated with better mental health.[52] Regarding local heritage as 'special' could thus be part of the pathway connecting heritage to well-being.

## Strengths and limitations

Our research exhibited a number of strengths. We used a sample of adults from across the whole of England, weighted to ensure representativeness to the wider population. We explored associations between exposure to heritage, and to visits, and to mental health, within relatively small areas across the whole of England, while other papers did not incorporate spatial data or nation-wide analyses. We explored whether mental health benefits of 'built' heritage remained regardless of the presence of green space in the local neighbourhood, providing a unique strand to this research. Regarding limitations, this study was cross-sectional; we cannot assume that key variables demonstrate causal associations, and that findings are generalisable within other contexts. We acknowledge that use of LSOA-level analysis may lead to statistical bias from using arbitrarily classified units to report spatial patterning.[53] Our study may be limited in that we do not know whether UKHLS respondents visit heritage sites within their local area or further afield. We may be limited by the simplified nature of the UKHLS questions on visits to heritage, for example, those visiting 'A city or town with historic character', may have done so for another purpose, that is, for everyday activities related to employment, education, food shopping, etc. Unfortunately, more detailed heritage visits data that can be linked to spatial data is not available. Doing every-day activities within a heritage environment may be associated with better mental health, however researching this is beyond the scope of our current study. Our study explores spatial exposure to heritage only, and does not incorporate other factors, which may promote or inhibit accessibility, such as financial access, for example, admittance fees. Although heritage may be free to view. We acknowledge the potential for a small number of HAR sites to be missing from the at risk register; lack of resources may result in systematic surveys of different HAR types being undertaken at different times.[54] Additionally, we acknowledge that individuals with poorer mental health may visit heritage sites less frequently due to barriers relating to motivations and opportunities.

## Policy implications

Our findings of inequities in heritage exposure, as a barrier to heritage engagement, are directly relevant to the UK Government's levelling up agenda. The bill recommends ensuring protection to existing heritage, tackling disparities in heritage access and improving cultural investment outside London.[41] Going forward the levelling up agenda must address inequalities in heritage exposure as part of a pathway to improve societal mental health. Methods to increase exposure within areas with fewer heritage sites could include better options

for public transport, for example, subsidised transport to areas with better exposure.[36] Exposure could be improved via investment in heritage in need of reinvigoration, such as at risk sites, and such sites can be improved via heritage volunteering schemes which, additionally, provide various social, psychological and health benefits to volunteers.[55] Increasing heritage engagement opportunities must consider both geographical and sociocultural factors.[18] Policy objectives to promote heritage engagement across all cultural groups should seek to redress potentially lower levels of interest among specific groups, such as poorer households.[17] Heritage organisations could expand 'public outreach' activities to neighbourhoods and communities with lower exposure,[36] or with lower levels of heritage awareness, and compensate for fewer heritage assets in deprived areas, through increased awareness of what heritage *is* there.

## CONCLUSION

We demonstrated that neighbourhoods with higher levels of income deprivation in England had lower levels of heritage exposure. Exposure to local heritage increased the likelihood of visiting heritage in general, and exposure to any heritage, or built heritage specifically, was positively associated with mental health, but only among those who had visited heritage in the past year. Our findings indicate that both spatial exposure and engaging with heritage through visiting are key. There is much to be gained from improving the structure of exposure. Formulation of strategies for schemes to tackle inequality in exposure in poorer neighbourhoods is necessary. Schemes could include working with communities and areas to improve access, knowledge, and awareness of heritage, and providing investment in 'at risk' heritage and currently non-accessible sites. Such schemes could promote local heritage as 'special', or distinctive in a positive way, to improve heritage engagement, and ultimately provide benefits to mental health and well-being.

**Acknowledgements** We wish to thank Neil Guiden (Data & Analysis Manager) at Historic England for supplying spatial data. We wish to acknowledge the UKHLS participants for their involvement in the study and the UK Data Service for supplying all social survey data used.

**Contributors** LMacdonald, EG, LMonckton and RM proposed the research questions. LMacdonald is the guarantor, led the project, created GIS variables and drafted all versions of the article. NN undertook statistical analysis and interpretation. All authors critically revised the article, read, and approved final draft.

**Funding** LMacdonald, NN and RM are funded by the UK Medical Research Council (MRC) Places and Health Programme (MC_UU_00022/4) and the Chief Scientist Office (CSO) (SPHSU2019) at the MRC/CSO Social and Public Health Sciences Unit, University of Glasgow, UK.

**Competing interests** None declared.

**Patient and public involvement** Patients and/or the public were not involved in the design, or conduct, or reporting, or dissemination plans of this research.

**Patient consent for publication** Not applicable.

**Ethics approval** This study involves human participants and the University of Essex Ethics Committee has approved all data collection on Understanding Society.

The UKHLS main study gained consent for all data linkages (except to health records) and respondents aged 16 or over provided informed and written consent to participate. Participants gave informed consent to participate in the study before taking part.

**Provenance and peer review** Not commissioned; externally peer-reviewed.

**Data availability statement** Data may be obtained from a third party and are not publicly available.

**ORCID iDs**
Laura Macdonald http://orcid.org/0000-0002-0593-8079
Natalie Nicholls http://orcid.org/0000-0003-0745-7065
Eirini Gallou http://orcid.org/0000-0003-3353-516X
Richard Mitchell http://orcid.org/0000-0003-3827-7155

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
