## [Reviewer comments · BMJ Open]

ARTICLE DETAILS

TITLE (PROVISIONAL)	Is spatial exposure to heritage associated with visits to heritage and to mental health? A cross-sectional study using data from the UK Household Longitudinal Study (UKHLS).
AUTHORS	Macdonald, Laura; Nicholls, Natalie; Gallou, Eirini; Monckton, Linda; Mitchell, Richard

VERSION 1 – REVIEW

REVIEWER	Dickens, Geoffrey Northumbria University, Nursing Midwifery and Health
REVIEW RETURNED	25-Sep-2022

GENERAL COMMENTS	Title While I am sure it is unintended the title gives the impression that the study could be longitudinal. I suggest changing it to ensure it is clear that it is cross-sectional. There is at least one too many 'withs' I suggest 'Associations between heritage, heritage visits and mental health' Abstract I see no need for the RQ acronym in the Abstract or elsewhere. Setting: the general population is the sampling frame. The setting might more correctly online or telephone (whichever, you do not state). Conclusions: It is for the reader to decide if the research is valuable. Your study does not have explanatory power and you overclaim by using 'benefits' as this implies causation. Strengths and limitations: What do you mean by weighted here? That the results were weighted or that sampling was weighted? Introduction p.4 Line 33 Ref [2] I think 'shown to benefit' claims more than the cited study claims for itself. Please be cautious in making causal inferences where study designs do not allow it. p.5 Para commencing Line 9 You present no supporting evidence for any of these claims. Line 56 You do not need to include GHQ12 as a standalone acronym at this point. Page 6 Line 7 'could improve' You are investigating whether it is associated not a causal link. Ditto for psychological wellbeing in the next line. Line 10. Again, the 'valuable' claim is for the reader to decide. Line 15 onwards. Again, I see no value added by use of RQ. Line 32 Do you mean georeference? Line 40. You state median area and population but you don't say whether LSOAs are devised to be geographically or demographically homogeneous i.e. what is the dispersion around the median? Page 7 Line 10. Battlefields are few? Wikipedia informs me there are 60 registered in England and Scotland. There may be others in
---

	Wales and NI. You say they are in specific geographical locations but surely so are castles etc. Please provide a stronger justification for this exclusion. Line 38 Including golf courses but not other sports facilities seems slightly odd. People don't go for a wander on the golf course in great numbers, and I am sure many golf clubs would not want them to. Line 42 'A city or town with historic character'. Surely anyone living near to would answer yes to this, but the purpose of their visiting might have been somewhat more mundane e.g. go to work, shopping, school run etc. Line 59. It may have been used previously but do these cut offs actually reflect anything meaningful i.e. do they optimally predict later diagnosis of caseness?. You are also missing a 'used in' between previously and green. Page 9 Line 29 You appear to have run multivariate analyses including all potential covariates rather than doing bivariate testing first in order that the multivariate analysis are purposeful i.e. maximise parsimoniousness (see Hosmer and Lemeshow) Discussion This is on the whole far more balanced and tentative than the 'headlines' you offer in the Abstract. I hope you will ensure that all aspects of the MS are equally balanced.
--	---

REVIEWER	Mak, Hei Wan University College London, Department of Behavioural Science and Health
REVIEW RETURNED	25-Oct-2022

GENERAL COMMENTS	The article explored three research questions: (1) whether spatial exposure to heritage varied by area income deprivation, (ii) whether spatial exposure to heritage linked to visiting heritage, and (iii) whether spatial exposure to heritage linked to mental health. The study found that more deprived areas had fewer heritage sites per population, neighbourhood exposure to heritage was associated with heritage engagement, and exposure to heritage was not associated with residents' mental health. Yet, the combination of heritage exposure and heritage engagement was associated with better mental health. I have read this article with great interest. It is very well-written and clear. The analysis was rigorous with the use of multiple datasets. I have some suggestions for authors to consider.  1. I wonder why only income deprivation was considered, but not IMD as a whole or other deprivation domains? 2. Given that the analysis involved multiple datasets, it would be very helpful if authors could provide a sample flowchart to indicate how the final, analytical sample was arrived. This would also help provide the proportion of missingness. 3. For heritage data exposures (page 8, line 22), were these exposure variables mutually exclusive? E.g. did scheduled monuments belong to category "built heritage" or category "scheduled monuments"? 4. What was the scale for heritage visits and how was it treated in the analysis? 5. Please provide an explanation for why it was not appropriate to use GHQ-12 as a continuous outcome? 6. Results shown on page 12 line 26 where authors reported that "a higher percentage of land as heritage did not increase the probability
--

	of visits to heritage (results not shown)" – wasn't this contradicting to the finding which showed areas with a higher density of historic buildings were associated with greater odds of visiting heritage? 7. For the interaction analysis, was it more of a stratification analysis than an interaction analysis? 8. On page 14 line 28, could the authors provide a reference for "LSOA size increase with increasing wealth"? 9. I found some of the results a little contradictory – so did wealthier areas have less heritage sites per area on average? If so, why is that? Why using different measures for heritage exposure would give different results? What were the data/results trying to tell us? 10. I wonder if there was any association between heritage exposure and psychological distress before adjusting for heritage engagement? 11. On page 16 line 10, authors suggested that "those with built heritage in their local... Showed lower distress" – did it contradict the earlier findings that showed there was no association between exposure and mental distress? 12. The article may benefit from a thorough proofread as some sentences sound a little casual.
--	--

REVIEWER	Mulaga, Atupele N. Malawi University of Business and Applied Sciences, Mathematical Sciences
REVIEW RETURNED	22-Dec-2022

GENERAL COMMENTS	I thank the authors for this important work. Here are the comments that may help the authors to improve the manuscript especially on the methods (statistical analysis) and presentation of the results. While the statistical analysis are appropriate the authors have not clearly justified the use of those statistical analysis. Abstract Objectives  • In lines 9-14 pp.2, authors should remove RQ in every line where it has been used. Methods  • Line 28 pp.2 the subtitle should be revised to include exposure measures i.e. outcomes and exposure measures or the exposure variables in line 28-29 should be removed. Results  • Line 39 pp.2, authors should revise how ORs ,95% CI are reported to follow standard practice • Lines 39-43 pp.2, authors should consider reporting ORs (95%CI) other than predicted probabilities for consistency Introduction  • In lines 15-22 pp 5, authors need to remove RQ in every line where it has been used and rewrite those sentences. Methods  • In lines 31-54 pp.8 statistical analysis section, authors need to rewrite these lines without repeating the research questions i.e., authors need to rewrite the analysis and indicate why that analysis was conducted in that they will be no need to list the research question. • In lines 31-35 pp.8, authors indicate that they compared median densities of sites however in table 1 of the results the mean densities were reported. May the authors clarify this • The authors need to justify why they used Kruskal Wallis test to compare the mean densities 1000 population and per square kilometers and mean percentage land cover cross area income
---

deprivation quintiles other than ANOVA

- In lines 39-44 and lines 54-58 pp.8, the authors used multilevel logistic regression to determine the association between heritage exposure and heritage visits and mental health. The authors need to justify the multilevel logistic regression model is suitable for this analysis
- In line 44-45 pp.8 the authors need to specifically indicate what this primary sampling unit of the survey to make it clear to the reader. In addition, there is need for the authors to briefly describe the sampling that was used in the UKHLS, this can be briefly described in lines 33-40 pp.7 under the section on visits to the heritage. This description may also help readers to understand why the multilevel logistic regression model was used.
- In line 56 pp.8 , authors should use mental health other than GHQ (as binary outcome) since GHQ is simply a health questionnaire module with questions which were used to calculate the measure of mental health.
- In line 12 pp.9 there is need to specify the actual primary sampling unit see bullet point number 5 above
- Line 23 pp.9 should be revised to make it clear to the readers.

Minor

- In line 7 DCMS should be written in full
- Lines 25-27 pp.7 could be more clear if its indicated that you merged data on heritage to individual level data
- Line 59-60 is not clear should be revised

Results

- In lines 41-44 pp.9, the authors need to present the result on comparing mean number of historic parks/gardens by EIMD income clearly; the result compares historic parks/gardens across EIMD income and not the other way. (THIS SEEMS TO BE OK)
- In lines 44-46 , the authors justify why means were used instead of medians however the authors donot justify why the Kruskal Wallis for comparing means was used other than the ANOVA. May the authors justify this.
- Authors should provide the test statistic value for the Kruskal Wallis/ANOVA together with the p-values in the last row of tables 1 and 2
- The authors need to check the totals across the columns for tables 1,2,3 or can the authors explain why the totals across column on historic parks/gardens adds up to 0.84 and not 4.19? The authors should check if the figures are correct.
- The figures in lines 19-23 pp.10 are not clear.
- In lines 19-23 pp.11 the authors need to report (AOR=,95% CI=) as is common practice and remove the reference to the research question RQ in parenthesis.
- Table 4 should include all other variables that entered in the regression model such as sex, age , household type etc ; all other variables that were used in adjusting the OR for the model should be reported in table 4
- Authors need to give a more informative table title to table 4 e.g. "Estimation of the odds of visiting heritage sites by measures of exposure to heritage using multilevel logistic model" or any other title that is more informative.
- For consistency I suggest that the authors report the adjusted odds ratios than predicted probabilities In Table 5
- Table 5 is not clear. Lines 17-21 pp.9 of the statistical analysis suggests that the post hoc multiple comparison tests were

	performed for variables with more than 2 levels however the presentation of the results in lines 9-25 pp12 and in table 5 suggests that the multilevel logistic model was fit to a sample on non visitors and visitors and these results are being compared across the variables.Can the authors clarify please ? The presentation of the results in lines 9-25 is not clear and does not match what has been indicated in lines 54-58 pp 4 in the statistical analysis section?What about results for all residents as indicated in lines 54-58 pp4? I recommend that the results for table 5 should be performed again as indicated in lines 54-58 of the statistical analysis exploring whether heritage exposure is associated with mental health among all residents,visitors and non visitors.Table 5 should include another column for all residents.  • My understanding of the statistical analysis in lines 17-21 pp.9 is that post hoc analysis is applied on exposure variables such as heritage per 1000 population which have more than two levels however the presentation of the results in lines 9-25 seem to suggest that comparison was between the non-visitors and visitors. I strongly suggest you consider the analysis again • The authors are also reporting an interaction effect in lines 7-14 pp.12 however this interaction term is not included in the results in table 5. Authors should include all the interaction terms that were included in the model. Are all the main exposure variables heritage,heritage per 1000 population and heritage per area interacting with heritage visits ?these should be included in the model reported in table 5. • Results for table 5 should also include all other variables that were used in adjusting the model Minor  • The authors should begin by making reference to the table numbers before presenting the results. This makes it easier for the readers to follow. • The authors should indicate clearly in table 5 that the CI in parenthesis is a 95% CI Discussion  • Authors should remove RQ in line 11 pp.13 and in every line where RQ has been included • Results being discussed in lines 8-17 pp.15 are important and the authors should include them in the supplementary materials
--	---

VERSION 1 – AUTHOR RESPONSE

Reviewer: 1

Dr. Geoffrey Dickens, Northumbria University Comments to the Author:

Title

While I am sure it is unintended the title gives the impression that the study could be longitudinal. I suggest changing it to ensure it is clear that it is cross-sectional.

There is at least one too many 'withs' I suggest 'Associations between heritage, heritage visits and mental health'

Response: Changed title to 'Is spatial exposure to heritage associated with visits to heritage and to mental health? A cross-sectional study using data from the UK Household Longitudinal Study (UKHLS).'

Abstract

I see no need for the RQ acronym in the Abstract or elsewhere.

Response: RQ acronym removed from abstract and throughout manuscript.

Setting: the general population is the sampling frame. The setting might more correctly online or telephone (whichever, you do not state).

Response: Now includes how the data was collected, i.e., via face-to-face interview or online questionnaire.

Conclusions: It is for the reader to decide if the research is valuable. Your study does not have explanatory power and you overclaim by using 'benefits' as this implies causation.

Response: Have removed the word 'valuable'.

Strengths and limitations: What do you mean by weighted here? That the results were weighted or that sampling was weighted?

Response: Have clarified sentence to reflect sampling was weighted.

Introduction

p.4 Line 33 Ref [2] I think 'shown to benefit' claims more than the cited study claims for itself. Please be cautious in making causal inferences where study designs do not allow it.

p.5 Para commencing Line 9 You present no supporting evidence for any of these claims.

Line 56 You do not need to include GHQ12 as a standalone acronym at this point.

Response: 'Shown to benefit' changed to 'associated with better'. Have removed 'highly valuable', and 'GHQ12'.

Page 6 Line 7 'could improve' You are investigating whether it is associated not a causal link. Ditto for psychological wellbeing in the next line.

Response: 'Could improve' replaced with 'is related to'. Have replaced 'benefits' with 'associations with well-being'

Line 10. Again, the 'valuable' claim is for the reader to decide.

Line 15 onwards. Again, I see no value added by use of RQ.

Line 32 Do you mean georeference?

Response: Have removed 'valuable' and 'RQ' and corrected to georeference.

Line 40. You state median area and population but you don't say whether LSOAs are devised to be geographically or demographically homogeneous i.e. what is the dispersion around the median?

Response: LSOAs were created to be socially homogeneous (where possible) and consistent in population size. Their geographical size varies according to level of urban/rural (i.e. rural LSOAs geographically larger as population more spread out). We have added this information to the paper, and added standard deviation values to supplemental table 1.

Page 7 Line 10. Battlefields are few? Wikipedia informs me there are 60 registered in England and Scotland. There may be others in Wales and NI. You say they are in specific geographical locations but surely so are castles etc. Please provide a stronger justification for this exclusion.

Response: Historic England data captures 47 battlefields (6 at risk), in 0.1% of LSOAs in England, while each of the other three heritage categories include thousands of sites. These small numbers meant we were unable to do comparison across income quintiles meaningfully. This detail now included in the methods.

Line 38 Including golf courses but not other sports facilities seems slightly odd. People don't go for a wander on the golf course in great numbers, and I am sure many golf clubs would not want them to.

Response: We discussed inclusion of categories with our Historic England collaborators who were keen for us to include green space as 'natural heritage' within analyses. Our reasoning was that golf courses are usually large green spaces which are visually highly noticeable.

Although it varies, golf courses can be accessible for non-golfers; they are not used for golfing 24/7, and may contain publicly accessible footpaths. Golf courses are often surrounded by other green space, and in some parts of the country coastal walks, so proximal access is available. We now include this detail on inclusion in the methods section.

Line 42 'A city or town with historic character'. Surely anyone living near to would answer yes to this, but the purpose of their visiting might have been somewhat more mundane e.g. go to work, shopping, school run etc.

Response: This is true, we are somewhat limited by the general nature of the questions in UKHLS but, unfortunately, more detailed heritage visits data linkable to location/spatial data is not available. We cannot rule out, of course, that doing more mundane activities within a heritage environment may be associated with better mental health, researching this is beyond the scope of our current research. We now include this in the methods section under limitations.

Line 59. It may have been used previously but do these cut offs actually reflect anything meaningful i.e. do they optimally predict later diagnosis of caseness?. You are also missing a 'used in' between previously and green.

Response: GHQ case-ness has been clinically validated, is used as a screening tool, and the threshold of four is appropriate to represent minor psychiatric morbidity 'case-ness' for the UK population, as this low level protects sensitivity (Goldberg et al. 1998). We have made this clearer in the text. The typo has been fixed.

Page 9 Line 29

You appear to have run multivariate analyses including all potential covariates rather than doing bivariate testing first in order that the multivariate analysis are purposeful i.e. maximise parsimoniousness (see Hosmer and Lemeshow)

Response: Our variables were initially chosen based on known prior associations with heritage visits and/or mental health, from a literature review, and we went on to examine full model coefficients (excel workbook available on request) to affirm that all variables included were found to be essential. A brief explanation of this now added to the methods. It is worth mentioning that approaches to building multivariate models vary from discipline to discipline, with what is considered good practice in one branch of science, potentially frowned upon in others. We do recognise the value of multiple bivariate testing but this is often more helpful when it is unknown whether predictors and outcome are associated, and can result in Type 1 error.

Discussion This is on the whole far more balanced and tentative than the 'headlines' you offer in the Abstract. I hope you will ensure that all aspects of the MS are equally balanced.

Response: Many thanks for taking the time to provide useful comments and suggestions; hopefully you will now find the MS to be more balanced overall.

Reviewer: 2

Dr. Hei Wan Mak, University College London Comments to the Author: The article explored three research questions: (1) whether spatial exposure to heritage varied by area income deprivation, (ii) whether spatial exposure to heritage linked to visiting heritage, and (iii) whether spatial exposure to heritage linked to mental health. The study found that more deprived areas had fewer heritage sites per population, neighbourhood exposure to heritage was associated with heritage engagement, and exposure to heritage was not associated with residents' mental health. Yet, the combination of heritage exposure and heritage engagement was associated with better mental health.

I have read this article with great interest. It is very well-written and clear. The analysis was rigorous with the use of multiple datasets. I have some suggestions for authors to consider.

Response: Many thanks for taking the time to provide useful comments and suggestions.

1. I wonder why only income deprivation was considered, but not IMD as a whole or other deprivation domains?

Response: It is essential to use the income sub-domain as the full index includes access/barriers to local services and health sub-domains, so there might have been some circularity in investigating whether it was associated with access to heritage/green space and/or mental health. We have now included this in the methods (see Methods section, '(i) local area population, context and socio-economic situation' section).

2. Given that the analysis involved multiple datasets, it would be very helpful if authors could provide a sample flowchart to indicate how the final, analytical sample was arrived. This would also help provide the proportion of missingness.

Response: Flowchart created and included (see methods).

3. For heritage data exposures (page 8, line 22), were these exposure variables mutually exclusive? E.g. did scheduled monuments belong to category "built heritage" or category "scheduled monuments"?

Response: The term 'built heritage' is a term we created and refers to scheduled buildings and listed buildings in a combined category. We understand the lack of clarity here as scheduled monuments appears twice in the list in error. We have removed the second occurrence of scheduled monuments.

4. What was the scale for heritage visits and how was it treated in the analysis?

Response: we have added extra detail into the methods section (see (iv) individual-level visits to heritage, health, and socio-demographic covariates – heritage visits). The heritage visits variable is described in the statistical analysis as a binary variable (no visits versus one or more visits yearly).

5. Please provide an explanation for why it was not appropriate to use GHQ-12 as a continuous outcome?

Response: GHQ-12 is a bounded interval scale variable with a limited range of integer values therefore we chose a binary version of the variable. GHQ case-ness has been clinically validated, is used as a screening tool, and the threshold of four is appropriate to represent minor psychiatric morbidity 'case-ness' for the UK population, as this low level protects sensitivity (Goldberg et al. 1998). This case-ness threshold has also been used in previous green space and mental health research (Astell-Burt et al. 2014). Have edited text to explain this choice.

6. Results shown on page 12 line 26 where authors reported that "a higher percentage of land as heritage did not increase the probability of visits to heritage (results not shown)" – wasn't this contradicting to the finding which showed areas with a higher density of historic buildings were associated with greater odds of visiting heritage?

Response: The % of land as heritage and the density of sites measure two different aspects of exposure (are likely to be related but don't measure the same thing). In terms of mean % land as heritage only 2% of LSOAs were covered by heritage, therefore we decided to remove this measure from the paper (we use densities per population and per area only). Hopefully, this simplification will make findings clearer.

7. For the interaction analysis, was it more of a stratification analysis than an interaction analysis?

Response: It was indeed an interaction analysis, as the models were fit with an interaction term. Could it be that the presentation of the results as 'non-visitor' versus 'visitor' in table 4 made it appear as if it was stratification? This presentation of results was used to help the reader to understand the interaction result. We have now explained this further in the statistical analysis section of the methods.

8. On page 14 line 28, could the authors provide a reference for "LSOA size increase with increasing wealth"?

Response: This can be seen in supplemental table 1 where mean LSOA size increases from deprivation quintile 1 (most deprived) to quintiles 4/5 (less deprived). The table is now referred to in the text.

9. I found some of the results a little contradictory – so did wealthier areas have less heritage sites per area on average? If so, why is that? Why using different measures for heritage exposure would give different results? What were the data/results trying to tell us?

Response: Densities per population and per area measure different aspects of exposure; these two measures were used as LSOAs differ in population size and geographical size (as explained in methods). In simple terms the findings show that wealthier areas have more sites per population head but these are likely to be spread over a larger geographical area. This is discussed in the 2nd paragraph of the discussion; the wording has been edited for clarity.

10. I wonder if there was any association between heritage exposure and psychological distress before adjusting for heritage engagement?

Response: There was a simple chi-square association between exposure (all heritage) and mental health, with more cases of psychological distress occurring in areas with lower exposure ($p < 0.002$).

11. On page 16 line 10, authors suggested that “those with built heritage in their local.... Showed lower distress” – did it contradict the earlier findings that showed there was no association between exposure and mental distress?

Response: The line supports our findings (see results - “Amongst those with one or more built heritage sites ... visitors had a lower predicted probability of distress”. There was an association but only amongst visitors.

12. The article may benefit from a thorough proofread as some sentences sound a little casual.

Response: We have proofread the article thoroughly.

Reviewer: 3

Ms. Atupele N. Mulaga, Malawi University of Business and Applied Sciences Comments to the Author:

I thank the authors for this important work. Here are the comments that may help the authors to improve the manuscript especially on the methods (statistical analysis) and presentation of the results. While the statistical analysis are appropriate the authors have not clearly justified the use of those statistical analysis.

Response: Many thanks for taking the time to provide useful suggestions and comments.

Abstract

Objectives

- In lines 9-14 pp.2, authors should remove RQ in every line where it has been used.

Response: RQ removed

Methods

- Line 28 pp.2 the subtitle should be revised to include exposure measures i.e. outcomes and exposure measures or the exposure variables in line 28-29 should be removed.

Response: Exposure added.

Results

- Line 39 pp.2, authors should revise how ORs ,95% CI are reported to follow standard practice

Response: text edited to include 95% CIs.

- Lines 39-43 pp.2, authors should consider reporting ORs (95%CI) other than predicted probabilities for consistency

Response: Table 5 (now table 4) presents the interaction effects from the model. While the model does produce ORs as (exponentiated) model coefficients, these are not easy to interpret directly because the reader must simultaneously consider the main effect and the variation in

it described by the interaction term. To interpret model interaction effects, it is well established that best practice is to assess predicted results, in a tabular or graphical format, hence the predicted probabilities tabulated here.

Introduction

- In lines 15-22 pp 5, authors need to remove RQ in every line where it has been used and rewrite those sentences.

Response: RQ removed

Methods

- In lines 31-54 pp.8 statistical analysis section, authors need to rewrite these lines without repeating the research questions i.e., authors need to rewrite the analysis and indicate why that analysis was conducted in that they will be no need to list the research question.

Response: The research questions have been removed and text rewritten.

- In lines 31-35 pp.8, authors indicate that they compared median densities of sites however in table 1 of the results the mean densities were reported. May the authors clarify this

Response: Median densities tended to be 0 so mean values were included as an alternative. Table 1 has been edited to include median values and interquartile range (IQR).

- The authors need to justify why they used Kruskal Wallis test to compare the mean densities 1000 population and per square kilometers and mean percentage land cover cross area income deprivation quintiles other than ANOVA

Response: Kruskal wallis was used as the distribution of these sites indicated that the parametric comparisons were not appropriate. This detail has now been added.

- In lines 39-44 and lines 54-58 pp.8, the authors used multilevel logistic regression to determine the association between heritage exposure and heritage visits and mental health. The authors need to justify the multilevel logistic regression model is suitable for this analysis

Response: As indicated in the methods, the outcomes are binary and random effects are necessary due to sampling methods used to collect the data, therefore multilevel logistic regression was suitable for analysis.

This detail has been added to the statistical analysis.

- In line 44-45 pp.8 the authors need to specifically indicate what this primary sampling unit of the survey to make it clear to the reader. In addition, there is need for the authors to briefly describe the sampling that was used in the UKHLS, this can be briefly described in lines 33-40 pp.7 under the section on visits to the heritage. This description may also help readers to understand why the multilevel logistic regression model was used.

Response: We undertook a secondary analysis of data. We have added additional information on the sampling method used by UKHLS, and full details of their sampling and PSU are available within a report and web page now cited (see (iv) individual-level visits to heritage, health, and socio-demographic covariates in methods section, and statistical analysis section (paragraph 4)). Supplemental table 2 provides further information on the sample used in analysis.

- In line 56 pp.8, authors should use mental health other than GHQ (as binary outcome) since GHQ is simply a health questionnaire module with questions which were used to calculate the measure of mental health.

Response: This has been changed for clarity.

- In line 12 pp.9 there is need to specify the actual primary sampling unit see bullet point number 5 above

Response: See response above.

- Line 23 pp.9 should be revised to make it clear to the readers.

Response: An explanation of complete case approach is now included (i.e., statistical analysis includes only respondents for which there is no missing data on the variables of interest).

Minor

- In line 7 DCMS should be written in full

Response: In its first use (page 3 of the introduction) DCMS is written in full.

- Lines 25-27 pp.7 could be more clear if its indicated that you merged data on heritage to individual level data

Response: have edited for clarity.

- Line 59-60 is not clear should be revised

Response: Additional information about the GHQ variable and reasoning behind use of the binary variable have now been included.

Results

- In lines 41-44 pp.9, the authors need to present the result on comparing mean number of historic parks/gardens by EIMD income clearly; the result compares historic parks/gardens across EIMD income and not the other way. (THIS SEEMS TO BE OK)

Response: This has been rewritten for clarity “Though significant, there was very little variation in density of historic parks/gardens per area across the deprivation quintiles.”

- In lines 44-46, the authors justify why means were used instead of medians however the authors donot justify why the Kruskal Wallis for comparing means was used other than the ANOVA. May the authors justify this.

Response: Kruskal wallis was used as the distribution of these sites indicated that the parametric comparisons were not appropriate.

- Authors should provide the test statistic value for the Kruskal Wallis/ANOVA together with the p-values in the last row of tables 1 and 2

Response: These have been added.

- The authors need to check the totals across the columns for tables 1,2,3 or can the authors explain why the totals across column on historic parks/gardens adds up to 0.84 and not 4.19? The authors should check if the figures are correct.

Response: The totals in tables 1-3 refer to overall means and are not derived by summing means across category levels. To make this clearer, the column header ‘total’ has been changed to ‘Overall’.

- The figures in lines 19-23 pp.10 are not clear.

Response: Line rewritten for clarity (“Around 1.2% (n=4832) of heritage sites (i.e., 2098 listed buildings, 2639 scheduled monuments and 95 historic parks/gardens) were deemed as ‘Heritage at Risk’).

- In lines 19-23 pp.11 the authors need to report (AOR=,95% CI=) as is common practice and remove the reference to the research question RQ in parenthesis.

Response: 95% CIs added and RQ removed.

- Table 4 should include all other variables that entered in the regression model such as sex, age, household type etc ;all other variables that were used in adjusting the OR for the model should be reported in table 4

Response: The variables that were adjusted for have been moved from under the table to under the table’s title to be seen more clearly.

- Authors need to give a more informative table title to table 4 e.g. “Estimation of the odds of visiting heritage sites by measures of exposure to heritage using multilevel logistic model” or any other title that is more informative.

Response: The title of the table has been changed to ‘Adjusted odds of visiting heritage by measures of exposure to heritage’. Text in the preceding paragraph now states that the table ‘...shows the findings of the multilevel logistic regression analysis as adjusted odds of visiting heritage by exposure to heritage sites’.

- For consistency I suggest that the authors report the adjusted odds ratios than predicted probabilities In Table 5

Response: See results section above for response of preference for predicted probabilities.

- Table 5 is not clear. Lines 17-21 pp.9 of the statistical analysis suggests that the post hoc multiple comparison tests were performed for variables with more than 2 levels however the presentation of the results in lines 9-25 pp.12 and in table 5 suggests that the multilevel logistic model was fit to a sample on non visitors and visitors and these results are being compared across the variables. Can the authors clarify please ?

The presentation of the results in lines 9-25 is not clear and does not match what has been indicated in lines 54-58 pp 4 in the statistical analysis section? What about results for all residents as indicated in lines 54-58 pp.4? I recommend that the results for table 5 should be performed again as indicated in lines 54-58 of the statistical analysis exploring whether heritage exposure is associated with mental health among all residents, visitors and non visitors. Table 5 should include another column for all residents.

Response: All participants are resident in their LSOA, with visit indicating that they have visited a heritage site (no analysis by non-residents). The table presents the effects of the interaction analysis from the models; the predicted probabilities of poor mental health given exposure to heritage is combined with whether the person has engaged with heritage or not. Each group (visit/exposure) has its own overall p, and then when comparing levels within groups, an adjusted p, to indicate which levels were different to each other. In the table the adjusted p-values are marked by stars, and in the text further explained. We have clarified what is presented in table 5 (now table 4) and edited the explanation of the statistical analysis in the methods.

- My understanding of the statistical analysis in lines 17-21 pp.9 is that post hoc analysis is applied on exposure variables such as heritage per 1000 population which have more than two levels however the presentation of the results in lines 9-25 seem to suggest that comparison was between the non-visitors and visitors. I strongly suggest you consider the analysis again

Response: Post hoc testing is used on independent categorical variables with more than two levels, to see if there are any differences between the levels, in addition to, comparing combinations of groups in interactions. Where heritage exposure was in three levels, post hoc testing was applicable. Post hoc testing was only used in conjunction with visit/no visit categorisation when examining its interaction with exposure, where there are more than two possible combinations to assess. The edited statistical methods section should make methods clearer to readers.

- The authors are also reporting an interaction effect in lines 7-14 pp.12 however this interaction term is not included in the results in table 5. Authors should include all the interaction terms that were included in the model. Are all the main exposure variables heritage, heritage per 1000 population and heritage per area interacting with heritage visits ?these should be included in the model reported in table 5.

Response: Table 5 (now table 4) presents the interaction from the models, as predicted results for each combination of the interactions. This is necessary, to interpret interactions you must examine predicted results, as a table or graph to determine what is happening (rather than

look at the model coefficient term). The table also states that these are the results from the model after adjustment for all other terms, including the main effect of exposure and visits to heritage itself. The edited statistical methods section should make methods clearer.

- Results for table 5 should also include all other variables that were used in adjusting the model

Response: The variables that were adjusted for have been moved from under the table to under the table’s title to be seen more clearly.

Minor

- The authors should begin by making reference to the table numbers before presenting the results. This makes it easier for the readers to follow.

Response: This has been added.

- The authors should indicate clearly in table 5 that the CI in parenthesis is a 95% CI

Response: This has been added.

Discussion

- Authors should remove RQ in line 11 pp.13 and in every line where RQ has been included

Response: RQ removed.

- Results being discussed in lines 8-17 pp.15 are important and the authors should include them in the supplementary materials

Response: Values for this finding (‘... those with built heritage in their local area, who had visited heritage, showed lower distress’) are included in the results section. We cannot include additional supplemental data for the research in lines 12-17 as this is from someone else’s research which we include citations for.

VERSION 2 – REVIEW

REVIEWER	Dickens, Geoffrey Northumbria University, Nursing Midwifery and Health
REVIEW RETURNED	25-Jan-2023

GENERAL COMMENTS	Thank you for your careful and considered response to the reviewers' comments. My concerns have been adequately addressed through addition and clarification of information.
--

REVIEWER	Mak, Hei Wan University College London, Department of Behavioural Science and Health
REVIEW RETURNED	25-Jan-2023

GENERAL COMMENTS	Thank you, authors, for revising the paper. It has improved greatly although some areas of the manuscript require further clarification. 1. P.4 lines 9-10. “One problem with studies of both green space and heritage is that they do not often allow for each other”. The sentence is a little unclear. I suggest rephrasing it. 2. P.8 lines 7-8. “We did not know whether visits to heritage were in respondent’s LSOA or not.” This feels a little out of place. I would suggest moving this to the limitation section. 3. P.9 lines 5-11. “To explore ... were not appropriate” . The sentence is quite long. I would suggest breaking it down.
--

	4. P.9 lines 37-41. “Models included, and tested ... main effects only).” I found this sentence a bit unclear. An interaction analysis interacting between exposure and visits were performed, but I am unsure why non-significant interaction terms were removed and why the models were re-run with the main effects only? I also wonder whether the 3 interaction terms i.e. heritage*visitor, heritage per 1000 pop*visitor, and heritage per area*visitor were added to the model simultaneously or individually? 5. P.14 lines 38-41. “Neighbourhood exposure to heritage was ... residents’ mental health”. What is the difference between “neighbourhood exposure to heritage” and “physical presence of neighbourhood heritage”? 6. P.15 lines 15-16. “and those with the financial means will pay more to live in homes with historic features [38]”. Please elaborate why people who are more affluent would pay more to live in areas with historic features. It is possible that they also prefer living in metropolitan cities where historic features may be limited. 7. The study shows that wealthier areas have higher mean heritage sites per population - is it possible that richer areas receive more resources to maintain those sites? 8. P.17 line 25. “Although heritage may often be viewed without admittance”. The sentence needs to be rephrased. 9. P.17 lines 27-30. “We acknowledge the potential for missing HAR data; lack of resources may result in systematic surveys of different HAR types being undertaken at different times [54]” Did the authors mean some HAR might have been missed from the current analytical dataset? The current sentence sounds like no HAR data were included in the analytical dataset. 10. P.17 lines 30-31. “Additionally, we acknowledge that depressed individuals may be less likely to visit heritage due to mental ill health reducing their drive to engage with heritage.” I would rephrase the sentence to e.g. 'individuals with poorer mental health may visit heritage sites less frequently due to barriers relating to motivations and opportunities' Please also elaborate how this is related to the current study. 11. P.17 lines 43-44. “Methods to increase exposure could include the development within areas with fewer heritage sites” What types of development within the areas? 12. “Schemes which encourage volunteering to improve at risk assets provide various social, psychological and health benefits to volunteers [55].” This feels a little out of place. How is this related to the findings? 13. P.17 lines 59-60. “with lower levels of heritage awareness, and compensate for fewer heritage assets in deprived areas, through increased awareness of what is there.” Please be specific about what 'what' is referring to e.g. the health/social benefits of heritage? 14. P.18 line 6. “more income-deprived neighbourhoods.” Did the authors mean 'neighborhoods with higher levels of income deprivation'? 15. P.18 lines 20-21.” Such schemes could promote local heritage as ‘special’”. What does it mean by ‘special’?
--	---

REVIEWER	Mulaga, Atupele N. Malawi University of Business and Applied Sciences, Mathematical Sciences
REVIEW RETURNED	03-Feb-2023

GENERAL COMMENTS	Authors need to insert figure one for the flow chart summary of GIS and individual data. The figure is missing The authors have adequately addressed all the comments.
---

VERSION 2 – AUTHOR RESPONSE

Reviewer: 2

Dr. Hei Wan Mak, University College London Comments to the Author:

Thank you, authors, for revising the paper. It has improved greatly although some areas of the manuscript require further clarification.

Response: Thank you for taking the time to include further comments, we have addressed these, see responses below.

1. P.4 lines 9-10. "One problem with studies of both green space and heritage is that they do not often allow for each other". The sentence is a little unclear. I suggest rephrasing it.

Response: Have changed to "One problem with studies of both green space and heritage is that they do not often acknowledge the overlap between them, i.e., the potential for co-occurrence of green space and heritage in the same area." The rest of the paragraph goes on to explain this further.

2. P.8 lines 7-8. "We did not know whether visits to heritage were in respondent's LSOA or not." This feels a little out of place. I would suggest moving this to the limitation section.

Response: This has been removed here and added to the limitations section.

3. P.9 lines 5-11. "To explore ... were not appropriate". The sentence is quite long. I would suggest breaking it down.

Response: The sentence has been broken into two sentences.

4. P.9 lines 37-41. "Models included, and tested ... main effects only)." I found this sentence a bit unclear. An interaction analysis interacting between exposure and visits were performed, but I am unsure why non-significant interaction terms were removed and why the models were re-run with the main effects only? I also wonder whether the 3 interaction terms i.e. heritage*visitor, heritage per 1000 pop*visitor, and heritage per area*visitor were added to the model simultaneously or individually?

Response: including an interaction term that isn't significant over specifies the model. Generally, in modelling, we want to aim for the simplest overall model that contains the variables of interest and the selected confounders. The extra interaction term, if not contributing to the model, was removed - this simplifies a model as far as possible. The sentence has now been changed to the following for clarity "Initially, 'heritage exposure/visits' interactions were included in the models to investigate moderating effects on mental health, with different exposure type/visit interaction run in separate models. If on testing, an interaction was found non-significant, it was removed and models re-run with main effects only. "

5. P.14 lines 38-41. "Neighbourhood exposure to heritage was ... residents' mental health". What is the difference between "neighbourhood exposure to heritage" and "physical presence of neighbourhood heritage"?

Response: No difference, changed physical presence to 'exposure' for clarity and consistency.

6. P.15 lines 15-16. "and those with the financial means will pay more to live in homes with historic features [38]". Please elaborate why people who are more affluent would pay more to live in areas with historic features. It is possible that they also prefer living in metropolitan cities where historic features may be limited.

Response: Have changed 'will' to 'may' pay. This was a finding within the research cited (ref 38).

7. The study shows that wealthier areas have higher mean heritage sites per population - is it possible that richer areas receive more resources to maintain those sites?

Response: Yes, this could be true, as geographical distribution of funding/resources in unequal. Last line of paragraph 2 in the discussion edited to emphasis this point "It is conceivable that HAR's greater presence within lower income areas (and greater numbers of heritage sites (overall) per population within wealthier areas), is a result of inequity in the geographical distribution of funding; the Levelling Up paper reports the need for investment in culture and heritage outside London, and within poorer areas"

8. P.17 line 25. "Although heritage may often be viewed without admittance". The sentence needs to be rephrased.

Response: Have changed to "Although heritage may be free to view."

9. P.17 lines 27-30. "We acknowledge the potential for missing HAR data; lack of resources may result in systematic surveys of different HAR types being undertaken at different times [54]" Did the authors mean some HAR might have been missed from the current analytical dataset? The current sentence sounds like no HAR data were included in the analytical dataset.

Response: Sentence changed to "We acknowledge the potential for a small number of HAR sites to be missing from the at risk register" It means that some sites might have fallen into disrepair but due to sites being surveyed at different times they may not be in the official HAR register (yet).

10. P.17 lines 30-31. "Additionally, we acknowledge that depressed individuals may be less likely to visit heritage due to mental ill health reducing their drive to engage with heritage." I would rephrase the sentence to e.g. 'individuals with poorer mental health may visit heritage sites less frequently due to barriers relating to motivations and opportunities'

Response: Have changed to suggested text.

Please also elaborate how this is related to the current study.

11. P.17 lines 43-44. "Methods to increase exposure could include the development within areas with fewer heritage sites" What types of development within the areas?

Response: The development refers to better options for public transport e.g., subsidised transport. Have edited sentence for clarity. This is within the policy implications paragraph and is mentioned as a part of the solution to improve heritage exposure in areas with poorer exposure (as obviously heritage can't be newly 'built' as it is historic).

12. "Schemes which encourage volunteering to improve at risk assets provide various social, psychological and health benefits to volunteers [55]." This feels a little out of place. How is this related to the findings?

Response: Sentence changed to "Exposure could be improved via investment in heritage in need of reinvigoration, such as at risk sites, and such sites can be improved via heritage volunteering schemes which, additionally, provide various..." " This is mentioned as a part of the solution to improve at risk heritage (found more often in poorer areas).

13. P.17 lines 59-60. "with lower levels of heritage awareness, and compensate for fewer heritage assets in deprived areas, through increased awareness of what is there." Please be specific about what 'what' is referring to e.g. the health/social benefits of heritage?

Response: 'what is there' refers to what heritage is there, have added the word heritage.

14. P.18 line 6. "more income-deprived neighbourhoods." Did the authors mean 'neighborhoods with higher levels of income deprivation'?

Response: Wording changed to neighbourhoods with higher levels of income deprivation.

15. P.18 lines 20-21." Such schemes could promote local heritage as 'special'". What does it mean by 'special'?

Response: By 'special' we mean something out of the ordinary or distinctive in a positive manner. This has been clarified in the text.

Reviewer: 1

Dr. Geoffrey Dickens, Northumbria University Comments to the Author:
Thank you for your careful and considered response to the reviewers' comments. My concerns have been adequately addressed through addition and clarification of information.

Response: Thanks once again for your earlier comments and suggestions.

Reviewer: 3

Ms. Atupele N. Mulaga, Malawi University of Business and Applied Sciences Comments to the Author:

Authors need to insert figure one for the flow chart summary of GIS and individual data. The figure is missing. The authors have adequately addressed all the comments.

Response: The journal required figure 1 to be included as a separate pdf which was uploaded so it should be available to download if you still wished to view it. Thanks once again for your earlier comments and suggestions.

VERSION 3 – REVIEW

REVIEWER	Mak, Hei Wan University College London, Department of Behavioural Science and Health
REVIEW RETURNED	03-Mar-2023
GENERAL COMMENTS	Thank you, authors, for the clear responses to my comments. My concerns have been adequately addressed.